# Participation and Performance Trends in the Oldest 100-km Ultramarathon in the World

**DOI:** 10.3390/ijerph17051719

**Published:** 2020-03-06

**Authors:** Beat Knechtle, Volker Scheer, Pantelis Theodoros Nikolaidis, Caio Victor Sousa

**Affiliations:** 1Medbase St. Gallen Am Vadianplatz, 9001 St. Gallen, Switzerland; 2Institute of Primary Care, University of Zurich, 8091 Zurich, Switzerland; 3Ultra Sports Science Foundation, 69310 Pierre-Bénite, France; volkerscheer@yahoo.com; 4Health Science Department, Universidad a Distancia de Madrid (UDIMA), 28400 Collado Villalba, Madrid, Spain; 5Exercise Physiology Laboratory, Nikaia 18450, Greece; pademil@hotmail.com; 6Bouve College of Health Sciences, Northeastern University, Boston, MA 02115, USA; cvsousa89@gmail.com

**Keywords:** ultra-endurance, sex, running, nationality, motivation, age group

## Abstract

Participation and performance trends in ultramarathon running have been investigated for large datasets and long period of times with an increase in participants and an improvement in performance. However, the analysis of ultramarathons across many decades is missing. We analyzed these trends for 96,036 athletes (88,286 men and 7750 women) from 67 countries competing between 1956 and 2019 in ‘100 km Lauf Biel’ in Switzerland, the oldest 100-km ultramarathon in the world. More men than women participated in all years. The number of male participants reached a peak at around 1985 and a decline in participation occurred thereafter. Women started competing in 1962. Men were always faster than women and both women and men reduced their race times over years. After about 1985, both overall women and men and both female and male winners were not able to improve race times. For men, athletes from all age groups below the age of 49 years old reached a peak of participation in the 1980s, and showed a decrease since then. Regarding age groups, the decrease first started in age group 20–29 years, followed by 30–39, 40–49, 50–59, and 60–69 years. For athletes in age groups 70–79 and 80–89 years, no decrease occurred. For women, age group athletes in age groups 40–49, 50–59, and 60–69 years increased their participation, whereas age groups 20–29 and 30–39 peaked in the late 1980s and started to decrease or stabilize, respectively. Switzerland, Germany, and France were the countries with the highest numbers of participants throughout the history of the race. In men, race times increased after about 1990 for most nationalities; only runners from Germany seemed to stabilize their performance. In women, runners from Italy, France, and Austria improved their performance over the years. In summary, the analysis of the oldest 100-km ultramarathon in the world showed a decrease in participation and an impairment in performance in the last 60 years. These changes were due to a decrease in the number of male ultramarathoners in around the 1980s, where mainly the number of age group runners younger than 70 years decreased.

## 1. Introduction

An ultramarathon is defined as any running event longer than the marathon distance (42.195 km) or taking longer than 6 hours of duration [1]. Officially, ultramarathon running started in 1921 with the ‘Comrades Marathon’, the ultramarathon with the longest tradition and the highest number of finishers worldwide [2]. Since then, ultramarathon running has been gaining popularity [3,4,5] where both the number of races and participants increased in recent decades [3]. Ultramarathon races are generally held as time-limited events (e.g., from 6 h to 10 days) [6] or distance-limited events (e.g., 100 km, 100 miles) [3,5,7].

The increase in participants in the last decades was primarily due to female runners and male age group runners [3,5]. However, in recent years, the number of youth ultramarathoners younger than 19 years also increased [8,9]. The increase in the number of races and participants has mainly been documented by analyzing the trends by calendar years for different kinds of specific races such as 100 miles [3,5], 100 km [7], or 24-h [6] races. To date, only one study investigated the historic trends in ultramarathon running by analyzing participation trends in all 100-mile ultramarathon running competitions held in North America between 1977 and 2008 [3]. Investigations of longer time frames are missing. Similarly, the performance trends in ultramarathon running have been investigated for different kinds of races such as 100 miles [3], 100 km [7], or 24-h [6] races.

While large datasets, including numerous races of different distances have been investigated o identify participation and performance trends in ultramarathon running for distance-limited races such as 100-km ultramarathons [7,10] or 100-mile ultramarathons [10], and time-limited ultramarathons such as 24-h ultramarathons [11], one must be aware that ultra-endurance races can appear and disappear, can move from one place to another, and a specific discipline in ultra-endurance (e.g., ultra-triathlon) can have ups and downs regarding the number of races, participants, and finishers [12]. 

Taking this into account, exclusive analyses of these trends in the evaluation of one of the world’s oldest ultramarathon is missing. The ‘100 km Lauf Biel’ in Switzerland [13] has been held at the same location without interruption since 1959. Switzerland is a small country with around 8.5 million inhabitants [14] and is located in the middle of Europe. Although Switzerland is very small, it has a very active scene with endurance and ultra-endurance competitions such as half-marathons [15], city [16] and mountain [17] marathons, different ultramarathons such as mountain [18] and 24-hour ultramarathons [19], cross-country skiing events [20], long-distance swimming [21], and cycling [22,23] events and multi-sports events such as the oldest Ironman triathlon in Europe [24] and the Duathlon World Championship ‘Powerman Zofingen’ [25]. 

In these Swiss races, differences in participation and performance have been found. An increase in participants has been reported mainly due to an increase in female and age group athletes for the Duathlon World Championship [25], for ‘Ironman Switzerland’ [24], for half-marathoners but not for marathoners [16]. However, in recent decades, world-famous races with a pioneering character such as one of the first mountain bike ultramarathons in the world, the ‘Swiss Bike Masters’ [26], and the longest inline skating race, the Inline 111’ [27], have disappeared. Furthermore, a study investigating master athletes in the ‘100 km Lauf Biel’ showed a decrease in participation of younger age group runners [28]. In relation to the famous ‘Jungfrau Marathon’, a mountain marathon with worldwide participation that also hosts the World Championship in mountain marathon running, participation increased in age group runners, but performance was found to be impaired [29]. Obviously, for some races held in Switzerland, important changes took place in both participation and performance with a decrease in participation.

Regarding the ‘100 km Lauf Biel’, local media reported in recent years a decrease in participants. In 2002, the race director argued that changes in society, concurrence of other ultra-distance running races, and new and trendy sports disciplines such as triathlon and inline skating had led to a decrease in the number of participants [30]. This decrease in the 100-km ultramarathon led to the decision to offer both half-marathon and marathon races as well [31]. In 2015, the race director reported that the number of participants had continued to decrease, especially runners from Germany. However, the race saw more participation in recent decades from Germany than Switzerland, indicating that fewer local athletes were competing than in earlier years [32].

Recent studies investigated the participation and performance trends in 100-km ultramarathons held worldwide [7,33,34]. Between 1959 and 2016, most of the finishes were achieved by runners from Japan, Germany, Switzerland, France, Italy, and USA with more than 260,000 runners [7]. Between 1998 and 2011, the number of finishers from Japan, Germany, Italy, Poland, and USA increased exponentially during the studied period [33]; between 1960 and 2012, 100-km ultramarathoners became faster [34]. What is missing, however, is the knowledge of the participation and performance trends of the world’s oldest 100-km ultramarathon with a detailed analysis of different aspects such as nationality and participation, and performance of age group athletes. 

Therefore, the aim of this study was to investigate the participation and performance trends of the world’s oldest 100-km ultramarathon held in Biel, Switzerland. Based upon the above-mentioned findings regarding participation and performance for specific Swiss races and comments in newspapers, we expected to find a decrease in participation in this specific 100-km ultramarathon. We also expected to find changes in participation by nationality over calendar years.

## 2. Materials and Methods 

### 2.1. Ethical Approval

This study was approved by the Institutional Review Board of Kanton St. Gallen, Switzerland, with a waiver of the requirement for informed consent of the participants, as the study involved the analysis of publicly available data. 

### 2.2. The Race

‘100 km Lauf Biel’ is the oldest 100-km ultramarathon in the world and has been held for the past 60 years in the same place and without a break. Participants must be 16 years or older with no other restrictions or limitations. The race takes place annually on the first weekend in June, with all runners starting on Friday evening at 10:00 p.m. The race ends on Saturday at 07:00 p.m. with a time limit of 21 h. The weather conditions at that time of the year are normally favorable with air temperature of 15–20 °C at night. The fast finishers usually complete the race within ~7 h and arrive early in the morning at around sunrise on the following day. The athletes are allowed to be supported by a cyclist for additional food and clothing. The race is held as one large loop of 100 km with a total change in altitude of ~645 m. The organizer offers 17 aid stations offering food and beverages, including hypotonic sports drinks, tea, soup, caffeinated drinks, water, bananas, oranges, energy bars, and bread.

### 2.3. Dataset

Data were obtained from the official race website (www.100km.ch) in the specific section (www.100km.ch/index.php/de/newsletter/bestenliste-1) for 1959–2019. Available data were year of the race, first and family name, sex, age group, race time, nationality, and age of the athlete. Only official finishers were considered.

### 2.4. Data Analysis

Data were analyzed using SPSS 25.0 (IBM, IL, USA). The Kolmogorov–Smirnov test identified normal distribution of the data. Two-way ANOVAs were used to assess performance in four models (Sex × Calendar Year; Sex × Age-group, Sex × Nationality, Nationality × Calendar Year). Non-linear regressions were applied when plotting two variables (performance ratio, calendar year, nationality, age-group) for all analyses, except when plotting the male and female champions of the year. In that particular case, two linear regressions were performed for each sex considering two stages of the race: 1959–1984 and 1985–2018; this intersection was determined for being the years with the highest number of participants. Statistical significance below 0.05 was accepted for all tests.

## 3. Results

A total of 96,036 athletes (88,286 men and 7750 women) from 67 countries officially finished the race between 1956 and 2019. More men than women participated across all years. The number of male participants reached a peak at 1984 and a decline in participation started in 1985. Women started competing in 1962 (Figure 1A).

The men-to-women ratio (MWR) for participation decreased over the years with an increase in female and a decrease in male participation (Figure 1B). Switzerland, Germany, and France were the countries with the highest numbers of finishers throughout the history of the race. During this time, mainly runners from European countries started the race, while runners from non-European countries started very rarely (Figure 1C).

The performance analysis of men and women showed a significant time-effect (F_6058_ = 22.4; *p* < 0.001; η_p_^2^ = 0.958), sex-effect (F_1471_ = 235.7; *p* < 0.001; η_p_^2^ = 0.333) and interaction Time × Sex (F_5,795,917_ = 2.732; *p* < 0.001; η_p_^2^ = 0.002). Men on average were faster than women, and both women and men reduced their race times over the years. Both overall women and men (Figure 2A) and both male and female winners (Figure 2B) were not able to improve race times after about 1985. The performance ratio between men and women showed a positive slope throughout the years, indicating a reduction in the gap in performance between women and men (Figure 2C).

Figure 3 shows the participation trend of men and women by age group. For men, all age groups below the age of 49 years reached a peak of participation in the 1980s, and showed a decrease since then. The decrease started first in the 20–29 years age group, followed by 30–39 years, 40–49 years, 50–59 years, and 60–69 years. There was no decrease in the age groups of 70–79 years and 80–89 years. For women, age group athletes in 40–49, 50–59, and 60–69 years increased participation, whereas age groups 20–29 and 30–39 peaked in the late 1980s and started to decrease or stabilize, respectively. All other female age groups always had low inconsistent number of participants along the years.

The performance analysis of men and women by age group showed a significant age-effect (F = 301.5; *p* < 0.001; η_p_^2^ = 0.022) and sex-effect (F = 4.362; *p* = 0.037; η_p_^2^ = 0.000), but no interaction Age × Sex (F = 1.686; *p* = 0.107; η_p_^2^ = 0.000). Age-groups 30–39 and 40–49 were the fastest for both men and women (Figure 4A). Additionally, the performance ratio between men and women had the highest ratio at 70 years of age and older (Figure 4B). 

An analysis by age group with the fastest athletes in the 10-year age groups showed that all groups reduced their race time along the years for both men and women (Figure 5). The performance analysis by nationality showed a significant sex-effect (F_1225_ = 31.9; *p* < 0.001; η_p_^2^ = 0.124), nationality-effect (F_1514_ = 53.4; *p* < 0.001; η_p_^2^ = 0.983) and an interaction Sex × Nationality (F_1,396,006_ = 3.00; *p* < 0.001; η_p_^2^ = 0.000). The fastest men were from Spain and Poland and the fastest women from Germany and Poland (Figure 6). 

Considering only the top 10 men, the performance analysis with nationality throughout the years showed a significant nationality-effect (F = 1071; *p* < 0.001; η_p_^2^ = 0.651), time-effect (F = 43.1; *p* < 0.001; η_p_^2^ = 0.429) and interaction Nationality × Time (F = 6.99; *p* < 0.001; η_p_^2^ = 0.385) for men (Figure 7A). Similarly, women also showed a significant nationality-effect (F = 266.5; *p* < 0.001; η_p_^2^ = 0.541), time-effect (F = 27.1; *p* < 0.001; η_p_^2^ = 0.533) and interaction Nationality × Time (F = 2.57; *p* < 0.001; η_p_^2^ = 0.312) (Figure 7B). In men, race times increased after about 1990 for most nationalities; only runners from Germany seemed to stabilize their performance. In women, runners from Italy, France, and Austria improved their performance over the years.

## 4. Discussion

This study investigated the participation and performance trends in the world’s oldest 100-km ultramarathon. The main findings were: (1) the number of men increased from 1959 to reach a peak around 1985 with a continuous decline thereafter, (2) women started competing in 1962, (3) a decrease in male age group runners since around the 1980s, (4) performance decreased for all male runners where only runners from Germany were able to stabilize their performance, and (5) there was a reduction in the gap in performance between women and men over the years.

### 4.1. Decrease in Male Participation after 1984

An important finding was that male participation increased from 1959 to around 1984 and started to decrease in 1985 with considerably decreases thereafter. The decrease was mainly due to a slump in the number of runners in the age groups of 20–29 to 60–69 years, whereas the number of male runners older than 70 years seemed to stabilize. This is in contrast to different reports describing an increase in both female and male participation in the last decades for different sports disciplines, such as marathons [35], ultramarathons [33], and triathlons [36]. A potential explanation could be that athletes were attracted after the 1980s to other sports disciplines, such as the Ironman triathlon [37], ultra-triathlons [12], or duathlons [25]. Other explanations could be that the race became less attractive due to a concurrence of other ultramarathons held nearby, time of the year, entry fee, and/or a low recognition of ultramarathon running. For example, it has been shown that long-distance triathlon races such as ‘Ironman Hawaii’ held in the USA [36] or ‘Norseman Xtreme Triathlon’ held in Norway [38] attract athletes from all over the world.

Local reasons can also explain the decrease in participation, since a decrease in participation has also been documented for other sports disciplines held in Switzerland in the last decades. The number of participants in specific line skating races such as ‘Inline 111’ [27] and specific mountain bike cycling races such as ‘Swiss Bike Masters’ [26] also decreased. More studies are needed to investigate trends in participation in specific races and/or sports disciplines in specific regions and/or countries. A decrease in participation in a specific race and/or sports discipline might also have occurred in other countries or regions.

### 4.2. Initial Increase and Then Decrease in Performance

We found that both overall and annual winners in both male and female finishers initially improved their performance with a stabilization after about 1985. This seems to be a general trend when a new sports discipline is launched. Lepers investigated the performance trends in both top ten women and men competing in ‘Ironman Hawaii’ between 1981 and 2007. He found that overall performance rapidly decreased after the start of the race to stabilize around the late 1980s. Regarding split disciplines, small improvements were found only for running and cycling [37].

An important finding was that women decreased the performance gap, although the MWR decreased. In other terms, a selection of the fastest women must have occurred. In recent years, several studies have found a decrease in the gap between female and male ultramarathoners [39], but a decrease in the MWR has only been reported for marathoners [40]. The reduction in gap seems to be a problem of age [39,41]. In time-limited ultramarathons, women were able to reduce the gap in age groups where they had relatively high participation [41].

### 4.3. The Aspect of Nationality

Regarding nationality, male runners from Germany were the only men with stable performance over decades and female runners from Italy, France, and Austria improved their performance. There seems to be a difference in sex regarding performance. A potential explanation could be the difference in motivation between female and male ultramarathoners [42,43]. For marathoners, for example, elite marathoners from Kenya [44] and Ethiopia [45] run for money to support their families. In contrast, marathon runners from Poland [46] and Greece [47] were not running to earn money.

‘100 km Lauf Biel’ is also more attractive for Europeans and specifically for athletes from Central Europe, but not for ultramarathoners from the USA. This might be explained by the large distance between the USA and Europe and by the fact that US-American ultramarathoners preferably compete in ultramarathons held in the USA and in races held in miles [3,5]. It is also possible that Europeans are being attracted to the novelty of other newer races nearby and choosing to compete in those instead of ‘100 km Lauf Biel’.

### 4.4. Sex Difference in Performance

We found that the performance ratio between men and women showed a positive slope, indicating a reduction in the gap between female and male performance over the years. Comparing overall winning and record performance times, men are faster than women at distances ranging from 100 m to 200 km [48], but women have closed the gap in some ultra-distance running events over the last 40 years [41]. A reduction of sex difference in performance has been shown for time-limited ultramarathons (i.e., 6, 12, 24, 48, 72, 144, and 240 h), where women were able to reduce the gap for most timed ultramarathons and for age groups where they had relatively high participation [41]. It has even been suggested that when matched for 50-km trail running performance, no sex differences were observed between performances in 80- and 161-km trail running events [49].

Possible explanations for this include the increase of female participating in endurance running events, training characteristics, and physiological aspects [41,50,51,52]. For example, in elite and recreational runners, the percentage of body fat is higher in women compared to men [53], but women may be better in utilization of fat for fuel in comparison to equally trained and nourished men [54], thus potentially providing them with an advantage for ultra-endurance distances [55]. It has also been shown that body fat and training characteristics were positively associated with running times in ultra-endurance events [28,56]. Running performance is also dependent on maximal oxygen uptake (VO_2_max) and running economy (RE), and although VO_2_max values are generally higher in elite man compared to elite women, no such sex differences were observed in RE [57,58].

There is evidence suggesting that women can exercise at higher proportions of VO_2_max during long ultra-endurance events than men, providing further explanation of why women are closing the gap in ultra-endurance running [59]. Psychological advantages may also influence performance and women may be better equipped and resilient to psychological stress and pain [60]. There are also sex differences in substrate metabolism during endurance exercise of moderate intensity, where women have a lower respiratory exchange ratio than men at the same relative intensity, indicating that they rely less on carbohydrate oxidation. In addition, women have a larger depot of intramyocellular lipid and greater part of intramyocellular lipids near the mitochondria, indicating that they have a greater capacity to use intramyocellular lipids [52].

Aging also seems to be different between women and men [61]. Although beyond the age of 65 years, the rate of aging shows a rapid increase, the rate of aging is slower in women compared to men [62]. There is also evidence for sex-specific variation in age-associated heart remodeling. There is evidence that the number of ventricular myocytes declines with age through apoptosis in men but not in women. This helps explain why older men are more likely than women to experience heart failure with reduced ejection fraction [63].

## 5. Conclusions

Although large studies investigating large datasets of ultramarathon races found an increase in participation and an improvement in performance in ultramarathon running, the analysis of one of the oldest ultramarathons showed a decrease in participation and an impairment in performance in the last 60 years. Both overall women and men and both male and female winners were not able to improve race times after approximately 1985. These changes were due to a decrease in the number of male ultramarathoners around the 1980s, where mainly the number of age group runners up to the age of 69 years decreased. However, the performance ratio between men and women showed a positive slope throughout the years, indicating a reduction in the gap between female and male performance. More studies are needed to investigate trends in participation in specific races and/or sports disciplines. Future studies need to investigate the participation and performance trends of other ultramarathons with a long tradition such as the ‘Comrades Marathon’. Furthermore, the trends in participation and performance by age groups should be investigated for all 100-km ultramarathons held in history. Future studies should also investigate what motivates female and male ultramarathoners from different countries.

## Figures and Tables

**Figure 1 ijerph-17-01719-f001:**
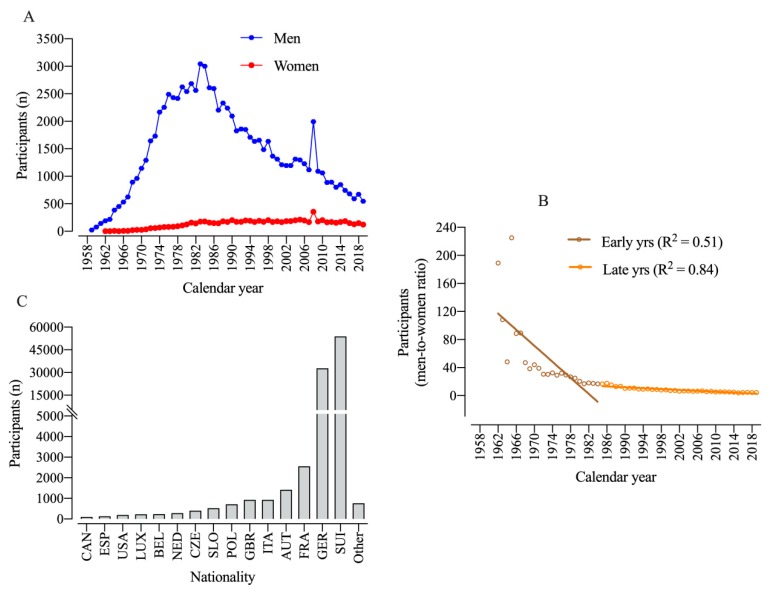
Participation of men and women in ‘100 km Lauf Biel’ over calendar years (**A**), men-to-women ratio (**B**), and participation of different countries (**C**). CAN = Canada, ESP = Spain, LUX = Luxembourg, BEL = Belgium, NED = Netherlands, CZE = Czech Republic, SLO = Slovenia, POL = Poland, GBR = Great Britain, ITA = Italy, AUT = Austria, FRA = France, GER = Germany, SUI = Switzerland.

**Figure 2 ijerph-17-01719-f002:**
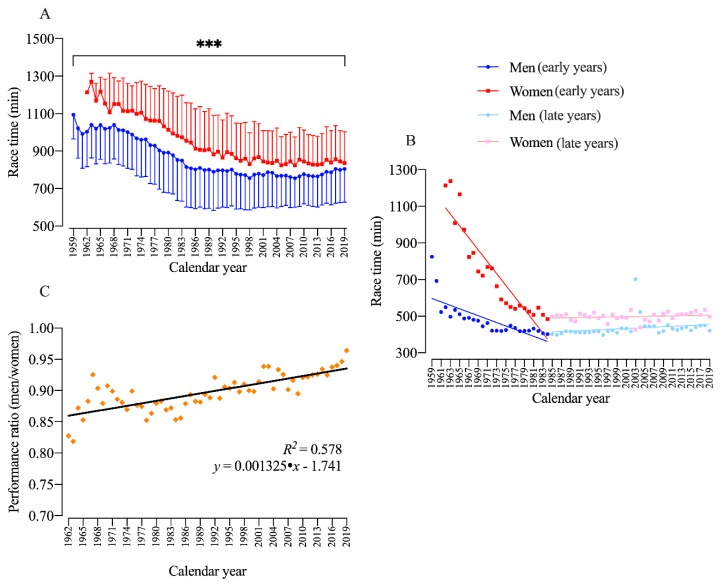
Performance of overall men and women (**A**) and annual winners (**B**) from 1959 to 2019; (**C**) Performance ratio during the same period. ***: sex, age and interaction effect (*p* < 0.05).

**Figure 3 ijerph-17-01719-f003:**
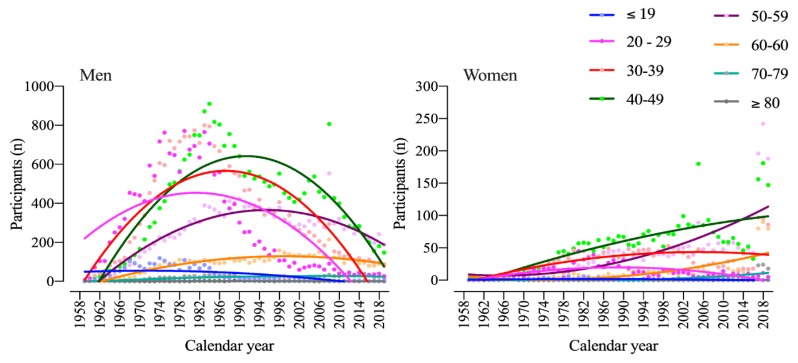
Participation by age group for men (**A**) and women (**B**) with second-order polynomial non-linear regressions.

**Figure 4 ijerph-17-01719-f004:**
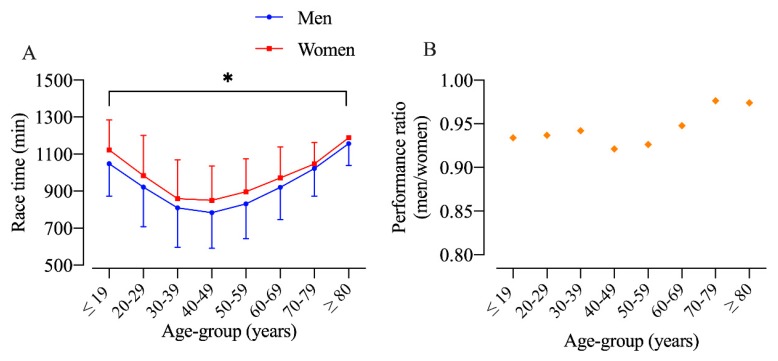
Performance of men and women in different age groups (**A**) and performance ratio (**B**) in the Biel ultramarathon. *: age-group effect (*p* < 0.05).

**Figure 5 ijerph-17-01719-f005:**
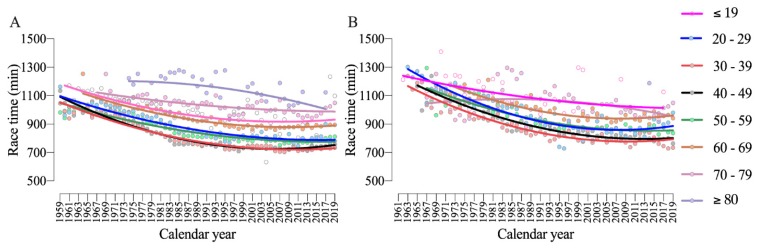
Performance trends of men (**A**) and women (**B**) by age groups with second-order polynomial non-linear regressions.

**Figure 6 ijerph-17-01719-f006:**
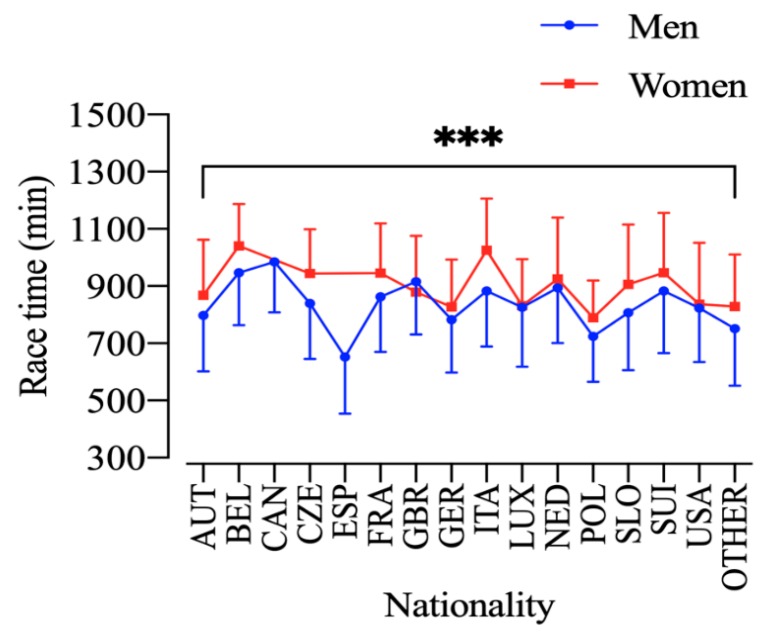
Performance of men and women in different countries. ***: sex, nationality and interaction effect (*p* < 0.05). AUT = Austria, BEL = Belgium, CAN = Canada, CZE = Czech Republic, ESP = Spain, FRA = France, GBR = Great Britain, GER = Germany, ITA = Italy, LUX = Luxembourg, NED = Netherlands, POL = Poland, SLO = Slovenia, SUI = Switzerland.

**Figure 7 ijerph-17-01719-f007:**
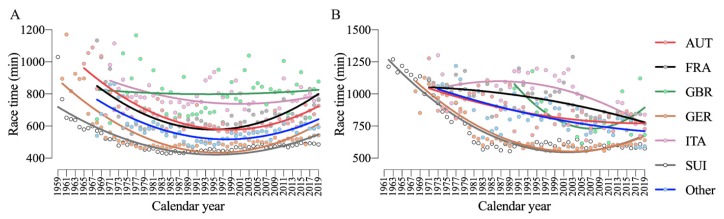
Performance trends of men (**A**) and women (**B**) by nationality throughout the years with second-order polynomial non-linear regressions. AUT = Austria, FRA = France, GBR = Great Britain, GER = Germany, ITA = Italy, SUI = Switzerland.

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
