# Peer review of "Participation and Performance Trends in the Oldest 100-km Ultramarathon in the World"

_ijerph, 2020, doi:10.3390/ijerph17051719_

Round 1

Reviewer 1 Report

See attached.

Author Response

General comments:

The manuscript provides a descriptive insight into the history of a 100-km ultramarathon. The unique dataset with such a long history of such an interesting event allows for interesting analyses. The authors provided a detailed overview of interesting aspects of the race (winning times, average finish times) for specific factors (race year, gender, gender ratio, age, country), but I am not convinced of the scientific contribution of the manuscript in its current state (particularly when keeping in mind (one of) the authors’ previous paper(s): “Age-related changes in 100-km ultra-marathon running performance”).

May 2 major concerns are that 1) the overload of analyses is not driven by any (theoretical) concept that the authors are trying to explain and 2) the analyses were done without too much attention to distinguish significant patterns from statistical significance due to large numbers. See below for more details.

My advice is to focus the narrative of the manuscript on one of the many components that the authors currently discuss and to do so with a more look at the ‘discovered’ patterns in the data. The most interesting angle to explore would be -in my opinion- that of the difference between men and women. Particularly given the recent media attention for this difference: Why women are outperforming men at the extremes of endurance (the Financial Times).

Answer: Unfortunately, we have no access to the newspaper article since it is not open access. We follow the recommendations of reviewer 3 to focus in the discussion also on sex difference.

Main concerns:

Trivial findings. Where does your hypothesis come from? Why are you not engaging in that specific hypothesis in the abstract? Why should we analyze the participation and performance trends of an ultramarathon? What insights might we gain from it?

Answer: We cite now newspaper articles from the last two decades where journalists mention that the participation decreased.

The fact that the number of participants, or even specifically the number of male participants increased and then decreased has no added scientific value. This looks more like an (incomplete) market research. Perhaps if the same analyses were done for other races (e.g., iron man), it would be somewhat interesting to see a shift in interest in a particular race or type of racing. These comparisons are now only made in the discussion. Although it is good that the authors make the link between other sports events, this link could have been made much more rigorously by actually comparing participation data (as most/some of these events also provide public data).

Answer: We think that we have not performed an incomplete market research.

The manuscript is lacking a rationale. Why are these analyses interesting? What can we learn from these analyses? By first formulating a concept of what the point the authors are trying to make, the analyses could be focused on a specific rationale (e.g., the difference in performance between men and women). By introducing each of the factors that are being analyzed, it gives the reader an idea as to why it might be interesting to look at these factors. Now, the different factors that are being examined come out of the blue. It feels as if the authors are just showing as many graphs as possible in the hope that one of them shows something interesting. I encourage the authors to bring some focus in their storyline.

Answer: We follow the recommendations of reviewer 3 to focus in the discussion also on sex difference.

Weak analyses. The selection of statistical models seems rather arbitrary. Why these (lines 94-95)? Many of the interpretations are made based on the fit that was made on each of the data points. With this number of datapoints it is easy to fit the data in many ways, but it seems that the authors have not at all examined which fit is actually the best fit. Look for example at Figure 1B. The fitted line suggest that the ratio is going back up again, whereas to the naked-eye, it seems that the trend continues to make the ratio more equal. When examining large datasets like this, I would expect some more caution with fitting the data. For example, De Leeuw and colleagues showed how the best polynomial can be chosen using cross-validation. Using this method, a much more reliable assessment of the robustness of a pattern can be made.

Answer: We investigated the selected variables as explained in the method section. Figure 1B has been adapted.

It seems to me that this manuscript is in an identity crisis. On the one hand, it tries to contribute to the understanding of participation and performance of a unique event, and on the other hand it tries to ‘mine the data’ for patterns that can be discovered in such a large dataset. For the first ‘identity’, the theoretical framework is far too weak. There are no hypotheses, there is no rationale why specific analyses might be interesting, there are no clear outcomes of the analyses that should have been considered as non-significant (i.e., not being able to reject the null hypothesis). For the second ‘identity’, the rigor in the analysis is simply not sufficient. There are no details available as to how the authors assured that their fits are the best fits possible for this dataset. I could come up with many other (statistically significant) ways to represent the data that would each have a (slightly) different narrative. To sum up, I find the topic fascinating and believe the manuscript could be interesting, but currently the scientific contribution of the paper is simply not good enough.

De Leeuw, A.-W. et al. (2018). Effects of pacing properties on performance in long-distance running. Big Data, 6(4). http://doi.org/10.1089/big.2018.0070

Answer: We think our study is no identity crisis. Thank you for the link to a paper from Big Data. However, we do not investigate pacing and the article is not open access so we do not consider this article.

Reviewer 2 Report

Written and organized well.  The premise of the study is noble and the underlying rationale for the comparisons is sensible.  In its present form, though, it appears to mix objective statistics with subjective speculations at times.  I had to study the graphs and re-read it several times to discern the two.  Specific examples are given below, but in general, the text associated with the discussion (and some subsections of results) should limit itself to what the data can objectively say (i.e., conclusions that anyone could make by doing their own analysis of the same data), and the text also needs to better explain what analyses were used for each data set and why those specific analyses were chosen.  

Major Comments:

The paragraph spanning lines 110-119 was confusing. Since this section underlies major points from the abstract and discussion, reworking would strengthen those points.  I understood that Figure 2-A represents all race participants, whereas Figure 2-B represents only the race winners, but after that it was confusing.  More specifically: Lines 21-22 in the abstract, and lines 112-113 in the text, say both sexes reduced their race times over the years, but then lines 116-117 contradict that.  Consider omitting the sentence in lines 116-117. The word “overall” is used differently throughout the paper, sometimes it refers to “all the race participants” (as in the caption to Figure 2-A) and other times it appears to refer to just a subset (as in line 116). It comes up again in the abstract (line 25) and in the discussion (line 213).  Please clarify how the word “overall” is being used in each instance. Many statements are based around 1985 being a turning point for men. Why/how was “1985” chosen as the year to emphasize?  Was there a statistical basis for that choice?  Later the manuscript switched to “1975-1985” in the discussion (lines 169, 174, and 212), but it was unclear why the switch was made.     I did not understand the claim that “women improved their race times throughout the whole period” with Figure 2-A (lines 113-114). If I look at Figure 2-A, overall women’s performance appears to plateau around 2005 (if not earlier).  If this claim is being made only because of Figure 2-C, then it should reference Figure 2-C and be moved later int eh text.  Or, if this claim is statistically justified based on Figure 2-A, then that justification should be explained. In lines 113-114, please specify the sentence is talking about all race participants. The reference to Figure 2-B should be moved from line 117 to line 116. The curves in Figure 2-B don’t seem to fit the data well. One option might be to set two linear equations to each data set, one for the men’s early years, one for the men’s later years, and the same for the women’s early and later years.  When you find the intersecting point for the men’s data, this might provide a better year to use for comment “c” above. Section 4.2 should be reconsidered: The emphasis in its first paragraph is a comparison to the Boston Marathon, but the Boston Marathon is only one race (and not necessarily representative of all racing events) so many generalizations from it is too tenuous. Example: Kathrine Switzer’s accomplishment is meritorious and should be celebrated, but women’s success in running is deeper than that (Violet Piercey’s IFAF-recognized marathon run in 1926, or even Marie-Louise Ledru’s 1918 run).  I acknowledge that Piercey’s and Ledrus’s runs weren’t large-group official marathons, but they prove a concept. The second paragraph is too speculative and should be removed altogether. Lines 126-127 say no decrease occurred in the men’s 70-79 and 80-89 age group, but line 170 says there’s been a decline in “male master age group runners.” What is meant by “master” in line 170?  If it’s an important concept, introduce it somewhere in the results and relate it to the data there first. Line 127 says that “for women, all age groups increased their participation”. Figure 3-C does not appear to support that claim.  There’s a purple line (I believe it’s women 20-29) that appears to peak around 1986 and then decrease after that.  The red line for women 30-39 also appears to be decreasing after 2005 or so.  If there’s a statistical basis for the claim, please provide it. In lines 138-139 and Figure 5, how were “fastest athletes” defined?

Minor Comments:

Line 100: It should be ’59 not ’56. Figures 3, 5, and 7 all state that the curves are generated by second-order polynomial non-linear regressions. In the Section 2.4, it says that “linear and non-linear regressions were also applied” (line 95), but it does not indicate which methods were used for which data.  Figures 1 and 2 do not indicate what method was used for their curves/lines (I’d suggest taking the curves out of Figure 2-B, as the one for women doesn’t fit the data well). Line 144, delete “and women”. Line 149 needs to reference Figure 7-B, because there’s no reference for it anywhere else. Line 219, first word should be “An” not “In”. Regarding the paragraph in lines 232-235: Is it possible Europeans are being attracted to the novelty of other newer races nearby, and choosing to compete in those instead of Lauf Biel? At line 225, please start a section 4.4, as the next two paragraphs correspond to your main idea “iv” from line 170, and all other main ideas from lines 167-172 had their own subsections.

Author Response

Comments and Suggestions for Authors

Written and organized well.  The premise of the study is noble and the underlying rationale for the comparisons is sensible.  In its present form, though, it appears to mix objective statistics with subjective speculations at times.  I had to study the graphs and re-read it several times to discern the two.  Specific examples are given below, but in general, the text associated with the discussion (and some subsections of results) should limit itself to what the data can objectively say (i.e., conclusions that anyone could make by doing their own analysis of the same data), and the text also needs to better explain what analyses were used for each data set and why those specific analyses were chosen.  

Answer: We agree with the expert reviewer and worked on the text and the figures to make the manuscript easier to follow.

Major Comments:

The paragraph spanning lines 110-119 was confusing. Since this section underlies major points from the abstract and discussion, reworking would strengthen those points.  I understood that Figure 2-A represents all race participants, whereas Figure 2-B represents only the race winners, but after that it was confusing. More specifically: Lines 21-22 in the abstract, and lines 112-113 in the text, say both sexes reduced their race times over the years, but then lines 116-117 contradict that. Consider omitting the sentence in lines 116-117.

Answer: We agree with the expert reviewer and deleted the sentence in lines 116-117.

The word “overall” is used differently throughout the paper, sometimes it refers to “all the race participants” (as in the caption to Figure 2-A) and other times it appears to refer to just a subset (as in line 116). It comes up again in the abstract (line 25) and in the discussion (line 213).  Please clarify how the word “overall” is being used in each instance.

Answer: We agree with the expert reviewer and deleted the sentences.

Many statements are based around 1985 being a turning point for men. Why/how was “1985” chosen as the year to emphasize?  Was there a statistical basis for that choice?  Later the manuscript switched to “1975-1985” in the discussion (lines 169, 174, and 212), but it was unclear why the switch was made.    

Answer: We agree with the expert reviewer and use the year 1985 consistently throughout the manuscript. This year was chosen by inspection of the raw numbers of male participants. The year of 1985 was the first year after the peak of participants in the race, one might even say that 1975-1985 were the “golden years” for this particular ultra-marathon due to its high popularity and never higher number of athletes. We agree that it may sound unclear the consistent appearance of this number throughout the manuscript and included an explanation in the results section.

I did not understand the claim that “women improved their race times throughout the whole period” with Figure 2-A (lines 113-114). If I look at Figure 2-A, overall women’s performance appears to plateau around 2005 (if not earlier).  If this claim is being made only because of Figure 2-C, then it should reference Figure 2-C and be moved later in the text.  Or, if this claim is statistically justified based on Figure 2-A, then that justification should be explained.

Answer: We agree with the expert reviewer and changed to ‘Both overall women and men (Figure 2-A) and both male and female winners (Figure 2-B) were not able to improve race times after about 1985’ to be consistent. We also changed in the abstract to ‘After about 1985, both overall women and men and both male and female winners were not able to improve race times after about 1985’.

In lines 113-114, please specify the sentence is talking about all race participants. The reference to Figure 2-B should be moved from line 117 to line 116. The curves in Figure 2-B don’t seem to fit the data well. One option might be to set two linear equations to each data set, one for the men’s early years, one for the men’s later years, and the same for the women’s early and later years. 

Answer: We agree with the expert reviewer and changed to ‘Both overall women and men (Figure 2-A) and both male and female winners (Figure 2-B) were not able to improve race times after about 1985’ to be consistent. The two linear trend lines for Figure 2-B were also added, as suggested.

When you find the intersecting point for the men’s data, this might provide a better year to use for comment “c” above.

Answer: We agree with both comments. We determined the “intersecting points” as the year of 1984-1985. Being 1984 the year of the peak number of participants in the race, and 1985 the first year of the beginning of a considerably decrease in the number of participants (and also performance according to the new analysis).

Section 4.2 should be reconsidered: The emphasis in its first paragraph is a comparison to the Boston Marathon, but the Boston Marathon is only one race (and not necessarily representative of all racing events) so many generalizations from it is too tenuous. Example: Kathrine Switzer’s accomplishment is meritorious and should be celebrated, but women’s success in running is deeper than that (Violet Piercey’s IFAF-recognized marathon run in 1926, or even Marie-Louise Ledru’s 1918 run).  I acknowledge that Piercey’s and Ledrus’s runs weren’t large-group official marathons, but they prove a concept.

Answer: We agree with the expert reviewer and deleted section 4.2

The second paragraph is too speculative and should be removed altogether. Lines 126-127 say no decrease occurred in the men’s 70-79 and 80-89 age group, but line 170 says there’s been a decline in “male master age group runners.” What is meant by “master” in line 170?  If it’s an important concept, introduce it somewhere in the results and relate it to the data there first. Line 127 says that “for women, all age groups increased their participation”. Figure 3-C does not appear to support that claim. There’s a purple line (I believe it is women 20-29) that appears to peak around 1986 and then decrease after that. The red line for women 30-39 also appears to be decreasing after 2005 or so.  If there’s a statistical basis for the claim, please provide it. In lines 138-139 and Figure 5, how were “fastest athletes” defined?

Answer: We agree with the expert reviewer. We believe the reviewer meant Figure 3-B instead of 3-C, since there is no 3-C. Anyhow, since we are dealing with raw numbers (not means and SDs) there is no statistical test to check for changes, the method is inspect the data and check for different numbers. Nevertheless, the reviewer is correct to suggest that different trends might have occurred for different age groups. Thus, we double checked the numbers and rephrased the results section mentioned in the comment. Please see changes in the manuscript marked as red.

Minor Comments:

Line 100: It should be ’59 not ’56. Figures 3, 5, and 7 all state that the curves are generated by second-order polynomial non-linear regressions. In the Section 2.4, it says that “linear and non-linear regressions were also applied” (line 95), but it does not indicate which methods were used for which data. 

Answer: We agree with the expert reviewer and changed the 2.4 section as following: “Non-linear regressions were applied when plotting two variables (performance ratio, calendar year, nationality, age-group) for all analyzes except when plotting the male and female champions of year, in that particular case, two linear regressions were performed for each sex considering two stages of the race: 1959-1984 and 1985-2018, this intersection was determined for being the years with the highest number of participants.”.

Figures 1 and 2 do not indicate what method was used for their curves/lines (I’d suggest taking the curves out of Figure 2-B, as the one for women doesn’t fit the data well).

Answer: We agree with the expert reviewer and attending to previous comments the method used for curves/lines were added in the 2.4 section, as well as a new analysis for Figure 2-B with two linear regression, instead of 1 non-linear.

Line 144, delete “and women”.

Answer: We agree with the expert reviewer and changed as requested.

Line 149 needs to reference Figure 7-B, because there’s no reference for it anywhere else. Answer: We agree with the expert reviewer and changed as requested.

Line 219, first word should be “An” not “In”.

Answer: We agree with the expert reviewer and changed as requested.

Regarding the paragraph in lines 232-235: Is it possible Europeans are being attracted to the novelty of other newer races nearby, and choosing to compete in those instead of Lauf Biel? Answer: We agree with the expert reviewer and added at the end of that section ‘It is also possible that Europeans are being attracted to the novelty of other newer races nearby, and choosing to compete in those instead of the ‘100 km Lauf Biel’.’

At line 225, please start a section 4.4, as the next two paragraphs correspond to your main idea “iv” from line 170, and all other main ideas from lines 167-172 had their own subsections.

Answer: We agree with the expert reviewer and started a new section ‘4.3. The aspect of nationality’

Reviewer 3 Report

Thank you for this article. I found this to be an interesting analysis of some of  the changing trends in ultra marathon running with an interesting take on sex, age and ethnicity comparisons relative to performance and over time. The figures were nicely done and easy to follow and see some of the conclusions that were made. My main suggestions come back to the discussion where I think some of the findings could be enriched by a greater research background on the physiological differences between men and women that affect endurance performance and the findings of the aptitude of females for ultra distance running due to fuel metabolism. I have provided a couple of links that may help in this are though there is other research out there. A similar discussion could be made about the sex categories and performance as there are natural biological declines with aging that of course explain performance trends as well.

Below are some specific suggestions:

Consider rewording the sentence on lines 64-65 for clarity.

Consider rewording the sentence on lines 105-106 for clarity.

Line 191-192 where the conclusion for a decline in ultramarathon enrollment locally was potentially a decrease in hype or boom could be better worded and explained most robustly by citing evidence of fitness trend changes.

Section 4.2. that discusses the changes in female times could be better supported with the use of the evidence regarding sex based physiological differences in substrate metabolism and how that supports females to be well positioned for long distance racing despite being slower than males for shorter distances such as the marathon. While the historical perspective is very interesting I think there is more that can be said here in terms of womens times improving, changes in womens training in sport and the biological and psychological aptitude that research has shown to give women an advantage in ultra long distance running.

I think this would also support the statements in section 4.3. Below are a couple of journals which delve more into the physiological differences between men and women and why women may have more comparable times to men in ultra running.

https://journals.lww.com/acsm-msse/Fulltext/2008/09000/Ultramarathon_Trail_Running_Comparison_of.18.aspx

https://springerplus.springeropen.com/articles/10.1186/s40064-016-2326-y

The conclusions should draw more attention to the improvement in womens times seen in this sport.

Author Response

Thank you for this article. I found this to be an interesting analysis of some of the changing trends in ultra-marathon running with an interesting take on sex, age and ethnicity comparisons relative to performance and over time. The figures were nicely done and easy to follow and see some of the conclusions that were made.

My main suggestions come back to the discussion where I think some of the findings could be enriched by a greater research background on the physiological differences between men and women that affect endurance performance and the findings of the aptitude of females for ultra-distance running due to fuel metabolism. I have provided a couple of links that may help in this area though there is other research out there.

Answer: We agree with the expert reviewer and expanded the discussion with the aspect of sex difference in performance and differences in fuel metabolism.

A similar discussion could be made about the sex categories and performance as there are natural biological declines with aging that of course explain performance trends as well.

Answer: We agree with the expert reviewer and added this aspect in the discussion.

Below are some specific suggestions:

Consider rewording the sentence on lines 64-65 for clarity.

Answer: We agree with the expert reviewer and changed to ‘The ‘100 km Lauf Biel’ (www.100km.ch) has been held at the same location without interruption since 1959’.

Consider rewording the sentence on lines 105-106 for clarity.

Answer: We agree with the expert reviewer and changed to ‘During this time, mainly runners from European countries started, while runners from non-European countries started very rarely’.

Line 191-192 where the conclusion for a decline in ultramarathon enrollment locally was potentially a decrease in hype or boom could be better worded and explained most robustly by citing evidence of fitness trend changes.

Answer: We agree with the expert reviewer and changed to ‘A decrease in participation in a specific race and/or sports discipline might also have occurred in other countries or regions.’

Section 4.2. that discusses the changes in female times could be better supported with the use of the evidence regarding sex based physiological differences in substrate metabolism and how that supports females to be well positioned for long distance racing despite being slower than males for shorter distances such as the marathon.

Answer: One of the other reviewers suggested to delete this section. We add now a new section about the sex difference in performance.

While the historical perspective is very interesting, I think there is more that can be said here in terms of women’s times improving, changes in women’s training in sport and the biological and psychological aptitude that research has shown to give women an advantage in ultra-long-distance running.

Answer: We agree with the expert reviewer and consider this aspect now in the discussion.

I think this would also support the statements in section 4.3. Below are a couple of journals which delve more into the physiological differences between men and women and why women may have more comparable times to men in ultra-running.

Answer: We agree with the expert reviewer and included these papers.

https://journals.lww.com/acsm-msse/Fulltext/2008/09000/Ultramarathon_Trail_Running_Comparison_of.18.aspx

https://www.ncbi.nlm.nih.gov/pubmed/18685521  

https://springerplus.springeropen.com/articles/10.1186/s40064-016-2326-y

https://doi.org/10.1186/s40064-016-2326-y

The conclusions should draw more attention to the improvement in women’s times seen in this sport.

Answer: We agree with the expert reviewer and add in the conclusions ‘However, the performance ratio between men and women showed a positive slope throughout the years, indicating a reduction in the gap between female and male performance over the years.’

Round 2

Reviewer 1 Report

Although I like the topic and I'm impressed with the many analyses that were done on the existing data, I struggle to see the scientific merit of this work because it doesn't seem to extend beyond describing the data.

I'm missing the story that the authors are trying to convey. What do these trends tell us? What should we learn from it?

Previously, I wrote that the findings seem trivial. Particularly because there is no clear rationale as to why these specific analyses (and not others) should be done.

I'm not saying that the work is not good, on the contrary I think it's rather interesting. But I believe that the storyline should be improved to 1) avoid being misleading and 2) take away the air of arbitrariness of the various analyses.

For example, the hypothesis is misleading:

"We hypothesized that changes in the trends of participation and performance over calendar years might be different to the worldwide trend when analysing combined data from several races and from several countries."

This suggests to me that a comparison will be made between multiple races from several countries to uncover any worldwide trends. The analysis in this manuscript, however, only focuses on one race, so this is very misleading.

As there is no such comparison between races, it feels arbitrary that the data is analysed the way it is. Why is the relation between year and participation examined? Why is the relation between gender and finish time examined? Why is performance examined per age category? What do we learn from the fact that participation to a specific race decrease over the last 60 years? What do we learn from the fact that there were fewer age group runners younger than 70 years?

In my opinion, the authors did a great job in writing a description of the trends of a specific ultramarathon, but not so much in writing a scientific paper that teaches us something new. For this paper to have any scientific contribution, the backdrop of each of these analyses should be properly introduced. Without a good framework, this paper is simply a collection of descriptions of a single event. Without a good framework, I don't know what we are supposed to learn from this paper.

In my opinion, a scientific paper has to portray some level of generalizability. In its current state, I don't see to what the findings of this paper generalize.

And finally, one specific comment:

Line 239: remove 'the'

Author Response

Comments and Suggestions for Authors

Although I like the topic and I'm impressed with the many analyses that were done on the existing data, I struggle to see the scientific merit of this work because it doesn't seem to extend beyond describing the data.

I'm missing the story that the authors are trying to convey. What do these trends tell us? What should we learn from it?

Answer: We expanded now the Introduction to show that some races held in Switzerland with pioneering character (Swiss Bike Masters, Inline 111) disappeared in the last decades. This leads to our hypothesis that the world’s oldest 100 km ultra-marathon held in Switzerland might suffer the same tragedy.

Previously, I wrote that the findings seem trivial. Particularly because there is no clear rationale as to why these specific analyses (and not others) should be done.

Answer: We explain now better in the Introduction that in several races held in Switzerland participation increased (female and master athletes). For some races, we mention that participation and performance (specific age groups) decreased. This leads to the idea to investigate the world’s oldest 100 km ultra-marathon held in Switzerland since several reports in the media clearly indicated a decrease in participation in the last decades.

I'm not saying that the work is not good, on the contrary I think it's rather interesting. But I believe that the storyline should be improved to 1) avoid being misleading and 2) take away the air of arbitrariness of the various analyses.

Answer: We add now the aspect of different races held in Switzerland in the last decades to show that participation increased in some races whereas in other races participation decreased dramatically so that several races disappeared although they had a pioneering character.

For example, the hypothesis is misleading:

"We hypothesized that changes in the trends of participation and performance over calendar years might be different to the worldwide trend when analysing combined data from several races and from several countries."

Answer: We add now some references to show that analyses of 100 km ultra-marathons held worldwide were done, but not for the world’s oldest 100 km ultra-marathon held in Switzerland

This suggests to me that a comparison will be made between multiple races from several countries to uncover any worldwide trends. The analysis in this manuscript, however, only focuses on one race, so this is very misleading.

Answer: We explain now better in the Introduction

As there is no such comparison between races, it feels arbitrary that the data is analysed the way it is. Why is the relation between year and participation examined? Why is the relation between gender and finish time examined? Why is performance examined per age category? What do we learn from the fact that participation to a specific race decrease over the last 60 years? What do we learn from the fact that there were fewer age group runners younger than 70 years?

Answer: As mentioned, in some races, participation and performance trends changed across calendar years. We explain now more in details what recent findings showed for Switzerland.

In my opinion, the authors did a great job in writing a description of the trends of a specific ultramarathon, but not so much in writing a scientific paper that teaches us something new. For this paper to have any scientific contribution, the backdrop of each of these analyses should be properly introduced. Without a good framework, this paper is simply a collection of descriptions of a single event. Without a good framework, I don't know what we are supposed to learn from this paper.

Answer: see comment below

In my opinion, a scientific paper has to portray some level of generalizability. In its current state, I don't see to what the findings of this paper generalize.

Answer: This is the third study to show that in Switzerland the participation decreases in a race after the disappearance of ‘Swiss Bike Masters’ and ‘Inline 111’, two pioneering races in ultra-endurance mountain biking and inline skating. Most probably the ‘100 km Lauf Biel’ will disappear in the next years when the actual trend continues.

And finally, one specific comment:

Line 239: remove 'the'

Answer: We deleted as suggested

Reviewer 2 Report

Thank you for the extensive and thoughtful revisions you undertook on this paper.  In light of the revisions, I'd like to make two very small, very minor requests (simply to polish off the paper):

Please rewrite line 27 of the abstract so it harmonizes with the revised lines 130-133. In Figure 2b, please add "early yrs" in parentheses after the darker symbols for "men" and "women" in the legend.

Author Response

Comments and Suggestions for Authors

Thank you for the extensive and thoughtful revisions you undertook on this paper.  In light of the revisions, I'd like to make two very small, very minor requests (simply to polish off the paper):

Please rewrite line 27 of the abstract so it harmonizes with the revised lines 130-133. In Figure 2b, please add "early yrs" in parentheses after the darker symbols for "men" and "women" in the legend.

Answer: Both Abstract and Figure 2b were corrected as suggested by the Reviewer.

Round 3

Reviewer 1 Report

Main Comments

My view on this manuscript is simple: the analyses are interesting, but they have not been embedded in a scientific narrative.

Consequently, there are no generalizable findings from this endeavour to describe a specific event (the 100 km Lauf Biel). It doesn't teach us anything about how trends for this one race are different compared to other races (as there is no direct comparison), let alone why this difference may exist. The current analyses tell us that a decrease in participation and a decrease in the gender gap exist, but it doesn't give us any insight as to why this may exist. This also not possible, given that the authors only analyse one dataset.

This manuscript is critically lacking an objective comparison. Despite their hypothesis about comparing trends worldwide, they do not compare (in their results) the trends from the 100 km Lauf Biel with any other race, instead, they only show descriptive results of this one specific race. The authors claim that their research is inspired by a trend for other races to disappear due to lack of participation, but their analyses do not support that story at all. Their analyses are about performance trends and gender differences (and so is their discussion). Although these performance trends are interesting (particularly the gender difference on these long endurance events), they have not been rigorously analysed with an a-priori hypothesis.

There is a severe disconnect with the different components of the paper:

Introduction -->

  • Participation in ultra races goes up and down (particularly in different age groups). Races might even disappear.
  • The 100 km Lauf Biel is unique for its long history.
  • Hypothesis: World wide trend might be different than trend for 100 km Lauf Biel

Methods/Results -->

Analysing ONLY the 100 km Lauf Biel:

  • Participation is going up and down over the years.
  • Many different nationalities participate.
  • Participation ratio men to women is become more balanced.
  • Race time has increased, but stabilized in the last decades.
  • Women are becoming more equal to men in terms of race time
  • And all of the above varies for different age groups

Discussion -->

  • Participation is going up and down because of popularity of other races
  • Performance is stagnating because of decreased popularity for 'fast' age-groups and nationalities.
  • Female performance continues to improve

The performed analyses do not align with the introduction. In the introduction, it is proposed that some worldwide trend is compared to a specific race, but in the analyses only the trends in one specific race are examined.

The disconnect also exists with the abstract. In the abstract, the authors talk about participation and performance going up and down, but don't mention anything about a comparison with a worldwide trend.

Although the methods/results and discussion seem quite well connected, it is impossible to judge how meaningful the findings are, as they are in the end not more than the observations of one single event. Therefore, it is nearly impossible to come up with any generalizable findings from this work. This leaves this work only as a description of a single event. And then my opinion about this work is straightforward: Although the results are insightful for this specific event (and perhaps their organizers), it doesn't have any scientific contribution as the results are in no way generalizable. 

In my opinion, I can see two versions of this paper that can provide a scientific contribution:

1) Comparison worldwide trends. Find similar datasets for other races, compare trends worldwide and show for which races the future is in danger.

--> Even with smaller datasets than the 100 km Lauf Biel dataset a comparison can be made. Such an addition would give the manuscript the necessary scientific rigor.

1) Development of rigorous pattern mining in race data. Requires a more rigorous assessment of when a pattern is a coincidence and when it is indeed a robust phenomenon in the data.

--> I'm not sure if this was the intention of the authors, but applying (and developing) 'data mining' techniques (that find robust patterns) to such datasets can be really interesting and novel.

Perhaps the authors can continue the narrative of the comparison of the trend at the 100 km Lauf Biel with other races (alternative 1). If they do pursue this direction, the abstract needs to be updated and the narrative should focus on 1) what is the trend worldwide? and 2) does the 100 km Lauf Biel stand out compared to other races? To address these questions, data of other races worldwide must be analysed in a similar fashion in order for this narrative to make any sense. Preferably, a comparison is made between races that seized to exist and races that thrived (as one of the motivations for this analysis was the fear that the 100 km Lauf Biel might disappear).

Specific Comments

Line 82: Please specify the direction of the changes (decrease in participation in this case, I presume).

Please look at the formatting of the newly written texts. Often, the spacing is inconsistent (e.g, line 70, no space between references 22 and 23) and there are some spelling mistakes (e.g., line 62, finishers instead of finisher).

Author Response

Main Comments

My view on this manuscript is simple: the analyses are interesting, but they have not been embedded in a scientific narrative.

Consequently, there are no generalizable findings from this endeavour to describe a specific event (the 100 km Lauf Biel). It doesn't teach us anything about how trends for this one race are different compared to other races (as there is no direct comparison), let alone why this difference may exist. The current analyses tell us that a decrease in participation and a decrease in the gender gap exist, but it doesn't give us any insight as to why this may exist. This also not possible, given that the authors only analyse one dataset.

This manuscript is critically lacking an objective comparison. Despite their hypothesis about comparing trends worldwide, they do not compare (in their results) the trends from the 100 km Lauf Biel with any other race, instead, they only show descriptive results of this one specific race. The authors claim that their research is inspired by a trend for other races to disappear due to lack of participation, but their analyses do not support that story at all. Their analyses are about performance trends and gender differences (and so is their discussion). Although these performance trends are interesting (particularly the gender difference on these long endurance events), they have not been rigorously analysed with an a-priori hypothesis.

There is a severe disconnect with the different components of the paper:

Introduction -->

  • Participation in ultra races goes up and down (particularly in different age groups). Races might even disappear.
  • The 100 km Lauf Biel is unique for its long history.
  • Hypothesis: World wide trend might be different than trend for 100 km Lauf BielAnalysing ONLY the 100 km Lauf Biel:
  • Methods/Results -->
  • Participation is going up and down over the years.
  • Many different nationalities participate.
  • Participation ratio men to women is become more balanced.
  • Race time has increased, but stabilized in the last decades.
  • Women are becoming more equal to men in terms of race time
  • And all of the above varies for different age groups
  • Discussion -->
  • Participation is going up and down because of popularity of other races
  • Performance is stagnating because of decreased popularity for 'fast' age-groups and nationalities.
  • Female performance continues to improveThe performed analyses do not align with the introduction. In the introduction, it is proposed that some worldwide trend is compared to a specific race, but in the analyses only the trends in one specific race are examined.Although the methods/results and discussion seem quite well connected, it is impossible to judge how meaningful the findings are, as they are in the end not more than the observations of one single event. Therefore, it is nearly impossible to come up with any generalizable findings from this work. This leaves this work only as a description of a single event. And then my opinion about this work is straightforward: Although the results are insightful for this specific event (and perhaps their organizers), it doesn't have any scientific contribution as the results are in no way generalizable. 1) Comparison worldwide trends. Find similar datasets for other races, compare trends worldwide and show for which races the future is in danger. 1) Development of rigorous pattern mining in race data. Requires a more rigorous assessment of when a pattern is a coincidence and when it is indeed a robust phenomenon in the data. Perhaps the authors can continue the narrative of the comparison of the trend at the 100 km Lauf Biel with other races (alternative 1). If they do pursue this direction, the abstract needs to be updated and the narrative should focus on 1) what is the trend worldwide? and 2) does the 100 km Lauf Biel stand out compared to other races? To address these questions, data of other races worldwide must be analysed in a similar fashion in order for this narrative to make any sense. Preferably, a comparison is made between races that seized to exist and races that thrived (as one of the motivations for this analysis was the fear that the 100 km Lauf Biel might disappear).As it is written in the title and the method section, the aim of the study was to analyze the participation and performance trends in the oldest 100 km ultra-marathon in the world.It makes also no sense to compare the oldest race in the world with all other races worldwide since they have a shorter tradition.Therefore, it is no option to compare the 100 km Lauf in Biel to any other race or to all races held worldwide since the aim of the study was clearly stated to analyze the oldest 100 km ultra-marathon in the world.
  • Others also analyzed only a single race with long tradition such as Med Sci Sports Exerc. 2008 Oct;40(10):1828-34, Age (Dordr). 2012 Jun;34(3):773-81, Open Access J Sports Med. 2012 Nov 2;3:169-74, Res Sports Med. 2020 Jan-Mar;28(1):121-137, Br J Sports Med. 2004 Aug;38(4):408-12 or Sports (Basel). 2019 Apr 12;7(4).
  • We expect that 500’000 to 1’000’000 runners completed a 100 km ultra-marathon since the first 100 km ultra-marathon in history. Obtaining and analyzing these data would take months and would be a new paper.
  • It is not possible to compare this race to any other race since it is the oldest in the world. Selecting any 100 km ultra-marathon in the world would lead to a serious selection bias because all other races have not this long tradition as the 100 km Lauf in Biel, Switzerland.
  • Answer: We thank the expert reviewer for his/her unbroken interest in our study.
  • --> I'm not sure if this was the intention of the authors, but applying (and developing) 'data mining' techniques (that find robust patterns) to such datasets can be really interesting and novel.
  • --> Even with smaller datasets than the 100 km Lauf Biel dataset a comparison can be made. Such an addition would give the manuscript the necessary scientific rigor.
  •  In my opinion, I can see two versions of this paper that can provide a scientific contribution:
  • The disconnect also exists with the abstract. In the abstract, the authors talk about participation and performance going up and down, but don't mention anything about a comparison with a worldwide trend.
  •  

We added in the Introduction ‘Recent studies investigated the participation and performance trends in 100 km ultra-marathons held worldwide [7,33,34]. Between 1959 and 2016, most of the finishes were achieved by runners from Japan, Germany, Switzerland, France, Italy and USA with more than 260'000 runners [7]. Between 1998 and 2011, the number of finishers from Japan, Germany, Italy, Poland and the United States of America increased exponentially during the studied period [33] and between 1960 and 2012, 100 km ultramarathoners became faster [34]. What is missing, however, is the knowledge of the participation and performance trends of the world’s oldest 100 km ultra-marathon with a detailed analysis of different aspects such as nationality and participation and performance of age group athletes to make the importance of the study clearer.

We adapted the hypotheses by changing that section to ‘Therefore, the aim of this study was to investigate the participation and performance trends of the world's oldest 100 km ultramarathon held in Biel, Switzerland. Based upon the above-mentioned findings regarding participation and performance for specific Swiss races and the comments in the newspapers, we expected to find a decrease in participation in this specific 100 km ultra-marathon. We also expected changes in the participation by nationality over calendar years.

In the conclusion, we added ‘Furthermore, the trends in participation and performance by age groups might be investigated for all 100 km ultra-marathons held in history’.

Specific Comments

Line 82: Please specify the direction of the changes (decrease in participation in this case, I presume).

Answer: We changed as suggested

Please look at the formatting of the newly written texts. Often, the spacing is inconsistent (e.g, line 70, no space between references 22 and 23) and there are some spelling mistakes (e.g., line 62, finishers instead of finisher).

Answer: We changed as suggested